# ZnO Nanoparticles Enhance the Antimicrobial Properties of Two-Sided-Coated Cotton Textile

**DOI:** 10.3390/nano14151264

**Published:** 2024-07-28

**Authors:** Agnė Giedraitienė, Modestas Ružauskas, Rita Šiugždinienė, Simona Tučkutė, Kastytis Grigonis, Darius Milčius

**Affiliations:** 1Institute of Microbiology and Virology, Faculty of Veterinary Medicine, Lithuanian University of Health Sciences, Mickeviciaus Str. 9, LT-44307 Kaunas, Lithuania; modestas.ruzauskas@lsmu.lt (M.R.); rita.siugzdiniene@lsmu.lt (R.Š.); 2Center for Hydrogen Energy Technologies, Lithuanian Energy Institute, 44403 Kaunas, Lithuania; simona.tuckute@lei.lt; 3Visdenta Ltd., LT-50185 Kaunas, Lithuania; kastytis@visdenta.lt

**Keywords:** nanoparticles, zinc oxide, two-sided-coated cotton textile, antimicrobial activity

## Abstract

Cotton textiles improved with metal oxide nanoparticles acquire additional features that may enhance their action against antimicrobial-resistant pathogens due to the unique properties and characteristics of the nanoparticles. The main objective of this work is to evaluate the antimicrobial features of two-sided-coated cotton textiles with ZnO nanoparticles. Nanoparticles were deposited using green chemistry technology with low-temperature oxygen plasma. ZnO particles formed stable structures on textile fibers. The optimal deposition parameters (150 W plasma power, 120 min immersion time) achieved the best effects against Gram-negative and Gram-positive bacteria and microscopic fungi. Two-sided-coated cotton with ZnO nanoparticles showed high antibacterial action on Gram-negative and Gram-positive bacteria. Modification with zinc oxide inhibited the growth of *Candida albicans* by more than half.

## 1. Introduction

Nanotechnology is advancing astonishingly, and industrial, mercantile, user, and healthcare products containing nanomaterials have appeared on the market [1]. Textiles functionalized with nanoparticles are used for clothing in hospitals, laboratories, wound dressings, filters, etc. [2,3]. Cellulose-based cotton textiles are trendy in the clothing industry because they are cheap, comfortable, and have good mechanical properties. One of the essential requirements for textiles is the comfort of wearing clothes. Cotton is the perfect material because it absorbs moisture and is breathable [4,5,6]. However, due to the large specific surface area and hydrophilicity of cotton, a favorable environment is created for the development of pathogenic microorganisms such as bacteria and fungi, which may cause unpleasant odors, allergic reactions, endanger hygiene and health, and deteriorate the strength of textiles [7,8]. One of the most challenging biomedical problems of the 21st century is the increase in bacterial infections and bacterial resistance to antimicrobial agents, which the development of new antibacterial substances could solve [9]. It is known that nanoparticles of transition metals have antibacterial activity [10]. Nowadays, the fundamental processes that underlie the bactericidal effects of nanoparticles are bacterial membrane disruption, reactive oxygen species (ROS) formation, nanoparticle penetration through the bacterial cell membrane, and intracellular antibacterial effects, including interactions with DNA and proteins [11,12]. The antibacterial properties of nanoparticles have been demonstrated for both Gram-positive and Gram-negative bacteria, and it depends on many factors such as nanoparticle structure, shape, size, synthesis, stabilizer type, etc. [13,14,15,16]. Metal oxide nanoparticles known to exhibit the most effective antimicrobial properties are Ag_2_O [17,18,19,20], Ag_4_O_4_ [21], CuO [17,20], Cu_2_O [17,19,22], ZnO [17,18,20], TiO_2_ [18,19], and MgO [17,19].

Scientists and industry have recently sought new metal nanoparticles and methods for functionalizing textile surfaces. Zinc oxide nanoparticles (ZnO NPs) are the most promising inorganic materials with bactericidal effects found in the composition of pharmaceuticals, sanitary products, cosmetics, food packaging processes, and medical textiles. ZnO NPs are biologically safe, non-cytotoxic to human cells, and biocompatible materials that can be directly used in biomedicine [23,24].

However, the targets of ZnO nanoparticles of clinical relevance to bacteria are not fully understood [25]. It is known that ZnO NPs can inhibit bacterial growth by interacting with the surface of bacteria or entering inside the bacterial cells. This disrupts bacterial enzyme systems, displacing magnesium ions necessary for bacterial enzymatic activity [26] and, thus, exhibits a significant bactericidal effect [23]. ZnO nanoparticles exhibit excellent growth inhibition of several bacterial strains. Chemically or biologically synthesized ZnO nanoparticles exhibited effective antibacterial properties against Gram-positive and Gram-negative bacteria and these clinical isolates: *Escherichia coli*, *Pseudomonas aeruginosa*, *Bacillus subtilis*, *Klebsiella pneumoniae*, *Salmonella typhi*, and *Staphylococcus aureus*. It was reported that ZnO nanoparticles synthesized by the green method showed a higher inhibitory effect than those synthesized by chemical methods [27].

To achieve effective nanoparticle stability on the fabric surface, the cellulose-based fabric must first be pre-activated or pre-treated. To develop durable, functional textiles, some bio-adhesive chemicals or methods that are harmless to human skin must be explored. Preventing agglomeration, desirable morphology, and uniform nanoparticle size have always been challenging in nanoparticle-related research. To overcome such challenges, ligand capping is commonly used [28].

Various synthesis methods, such as sol–gel, hydrothermal, CVD, etc., were used for ZnO NP deposition on textiles and other substrates. It is widely used for photocatalytic, photovoltaic, and other applications. These methods are expensive, time-consuming, and, in some cases, inefficient regarding material usage. They require large amounts of energy to operate, and most include several steps of ZnO deposition [29,30]. Various chemical substances are used in sol–gel, and CVD, which could harm the environment [31,32].

On the contrary, textile modification using physical vapor deposition (PVD) methods, such as magnetron sputtering, usually involves Zn or ZnO substrates and oxygen and argon as working gases. The deposit can be achieved through a technological procedure, and no toxic substances are generally released into the environment [33,34]. If green electricity is used for deposition procedures, PVD could be considered an ecological and waste-free nanoparticle production technology. However, due to high initial investment requirements related to the need for high vacuum systems, PVD methods could also be expensive. Glow discharge sputtering was used in this work to simplify requirements and expenses for the deposition systems. It uses energy input to initiate surface diffusion at the substrate and, thus, to form dense and well-adherent nanoclusters or coatings at low substrate temperatures and low vacuum conditions [35]. Glow discharge plasma offers several significant benefits for developing and functionalizing nanoparticles on textile surfaces, making it a valuable tool in textile engineering and nanotechnology. The textile surface is continuously bombarded by different particles (for example, working gasses ions) arriving from plasma and influenced by UV radiation during deposition. It can positively modify the surface properties by increasing wettability and surface energy. This improved surface activation enhances the adhesion of nanoparticles to the textile fibers. Plasma treatment also introduces various functional groups (such as carboxyl, hydroxyl, and amino groups) onto the textile surface, providing nanoparticle anchor points and leading to better attachment and uniform distribution. This type of treatment can be performed without additional chemical agents or solvents, making the process more environmentally friendly and reducing the potential for chemical waste. Plasma parameters (such as power, pressure, and gas composition) can be finely tuned to control the deposition and distribution of nanoparticles on textile surfaces. This deposition method can be utilized to deposit nanoparticles layer by layer, allowing for precise control over the thickness and composition of the coating.

In this work, both sides of cotton fabric were treated with ZnO metal oxide nanoparticles. The fabric was immersed in low-temperature plasma. Zn cathodes were used as sources of Zn particles, and oxygen was used as a working gas. The pulsed DC was used for plasma initiation, and deposition at a wide power range was investigated (the range from 20 to 220 W); the synthesis time varied from 15 to 120 min. The primary purpose of this work was to assess the antimicrobial activity of cotton fabric two-sided-coated with ZnO 150 W on 16 microbial strains (Gram-positive and Gram-negative bacteria and fungi).

## 2. Materials and Methods

### 2.1. Preparation of Samples

Textile samples (100% cotton, density—138 gsm), with geometry 10 cm × 10 cm, were used for ZnO deposition. Firstly, the unmodified cotton textile samples were autoclaved. Next, the deposition of ZnO NP particles using Zn cathodes (99.9% purity, manufactured by KJLC, Clairton, PA, USA) in a low-temperature plasma was performed, as described in detail in our previous published work [36]. A pulsed direct current (20 W to 220 W) was applied to generate the low-temperature plasma. Oxygen gas (purity 99.999%) was used as the working gas. The deposition processes were conducted at 10 Pa pressure.

ZnO nanoparticles were also deposited on glass samples in the same deposition system during a separate synthesis process. The same deposition parameters were used for the textile substrate.

### 2.2. Characterization of Materials

Scanning electron microscopy was used for surface morphology investigations (SEM Hitachi S-3400N, Tokyo, Japan), and all measurements were conducted at 5.0 kV in secondary electron mode. The energy-dispersive X-ray spectroscopy (EDS) technique (Bruker Quad 5040, Billerica, MA, USA) was employed for surface elemental composition analysis and elemental mapping of textile surfaces coated with ZnO nanoparticles. In non-contact mode, three-dimensional surface images were taken with the NT-206 atomic force microscope (AFM, NT-206, Microtestmachines, Gomel, Belarus). The surface topography results were also used to calculate the surfaces’ average roughness (Ra). For precise elemental analysis, X-ray photoelectron spectroscopy (XPS) measurements using the PHI VersaProbe spectrometer equipped with an Al monochromator, operating at 25 W beam power with a 100 µm beam size were conducted. The instrument’s energy scale was calibrated using the Au 4f7/2 peak at 84.0 eV and the Cu 2p3/2 peak at 932.6 eV. A dual charge neutralization system comprising low-energy electron and ion sources was employed to mitigate sample charging effects. Additionally, the adventitious carbon C1 component to 284.8 eV was aligned to compensate for the predominant charge effect. The X-ray diffractometer (XRD, Bruker D8 Discover Hamburg, Germany), Cu Kα radiation, and a Lynx Eye linear position detector at 2 theta angles from 20° to 70° were used for structural composition analysis.

### 2.3. Microbial Culture Collection

The microbial culture collection from the Microbiology and Virology Institute of the Lithuanian University of Health Sciences comprised 16 strains of reference, clinical, and zoonotic microorganisms, including Gram-negative and Gram-positive bacteria and yeasts, which were used in experimental activities [36].

### 2.4. Evaluation of Antibacterial Activity of Textile Samples Decorated with ZnO NPs

The antibacterial activity of textile two-sided-coated samples with ZnO NPs was evaluated by applying the methodology described in our previous published work [36].

### 2.5. Statistical Analysis

The received results were analyzed using the R statistical package (R-project.org; version 3.6.2) and considered more meaningful than *p* < 0.05. Graphics were performed using Microsoft^®^ Excel^®^ for Microsoft 365 MSO (Version 2312 Build 16.0.17126.20132) 64-bit.

## 3. Results and Discussion

### 3.1. Coating ZnO NPs on the Textile Substrates

ZnO nanoparticles were simultaneously deposited on both sides of the textile substrates. Figure 1 shows SEM morphology images of uncoated and coated substrates. Both sides of the cotton fabric were treated with ZnO metal oxide nanoparticles by immersion of the substrates in low-temperature plasma in an oxygen atmosphere at different power sources (in the range from 20 to 220 W), and synthesis time (from 15 to 120 min) was used in this work. A 150 W power source and 120 min deposition time were chosen as the best conditions for deposition because ZnO deposition was observed, and substrate materials maintained their integrity.

Analysis of the surface morphology of the uncoated textile substrates revealed homogeneous fibers (Figure 1a,c). After immersion in Zn-O low-temperature plasma, it became covered with uniform structures (Figure 1b,d). The elemental composition analysis presented in Table 1 confirmed the presence of carbon and oxygen in initial substrates and a small amount of Zn (up to 1.59 at. %) after deposition procedures.

Figure 2 represents the EDS mapping results of cotton fabric surfaces of initial (a,c) and ZnO-coated textile surfaces at 1.00k (b) and 3.00k (d) magnifications.

EDS mapping showed that most elements (C, O) were homogeneously distributed. Some uniformity could be observed in the case of Zn, which could be attributed to a small amount of deposited material and insufficient coverage of the substrate fibers’ surface.

### 3.2. Coating ZnO Nanoparticles on the Glass Slide

Glass slides were used as substrates to understand ZnO NP’s distribution on the surface. The synthesis parameters were used similarly to conditions related to textile substrates. The EDS results in Table 2 confirm the presence of Zn on the substrate surface. It is worth mentioning that concentrations of all other elements, such as Si, Mg, Ca, etc., stay almost the same, and it confirms the suggestion that ZnO forms a thin, uneven, modified layer on the glass surface.

Elemental mapping results are presented in Figure 3. It confirms that most of the elements were distributed uniformly except Zn. The presumption that ZnO does not form uniform structures due to the formation of cluster-like structures or insufficient amount of material to cover glass surface uniformly could be done.

Using AFM, the surface topography of the sample before and after the deposition of ZnO nanoparticles was investigated. The obtained results are presented in Figure 4.

Glass substrate surface analysis revealed that before deposition, it was smooth, accompanied by individual irregularities, which affected the relatively high surface roughness (R_a_ = 1.3 nm). After ZnO NP’s deposition (Figure 4b), the surface becomes covered by uniform, pyramid-like structures, which influenced the increase of surface roughness (R_a_ = 68 nm). ZnO structures could be attributed to the initial stages of growing ZnO thin film on the glass substrate surface.

The structural analysis of the deposited ZnO was performed based on X-ray diffraction measurements. The pattern of ZnO film is presented in Figure 5. The results showed a highly textured hexagonal wurtzite crystal structure (JCPDS card: 01-079-5604) in the sample with the most intense (002) plane. The wurtzite crystal plane (002) was observed at a 2θ angle of 34.4°. The crystallite size analysis was performed for the ZnO plane (002) using TOPAS-64 V6 software. The crystallite size observed for the sample was 3.2 nm. It is worth mentioning that ZnO deposited on both substrates (glass and textile) has only one peak at a 2θ angle of 34.4°. In the case of textiles, the ZnO peak mostly overlaps with the peak of the textile substrate; however, after ZnO deposition, the peak increases in intensity and slightly moves to the left, becoming like the peak on a glass substrate. The XRD results prove that ZnO crystalizes in the same fashion on both glass and textile substrates.

Elemental composition and of compounds in the near-surface regions were further investigated using the XPS technique. The survey spectrum of the Zn-O sample surface is presented in Figure 6 (red curve). Zn, O, and C elements were detected on the initial surface of the sample. To determine whether carbon is solely present on the sample surface or if it is also embedded within the sample film, surface sputtering using an Ar^+^ ion gun was conducted. Initially, two relatively mild sample sputtering steps (each lasting 1 min) with an acceleration voltage of 2 kV and a raster size of 2 mm × 2 mm were applied. After each sputtering step, a new survey spectrum was obtained. The same three elements were observed in both cases, but the carbon C 1s peak consistently decreased in intensity.

To explore the sample’s bulk composition, the ion gun’s acceleration voltage was increased to 4 kV, and two additional sputtering steps (each lasting 2 min) were performed with the same ion beam rastering. Comparative analysis of XPS spectra before and after all ion gun sputtering steps (as shown in Figure 1, represented by the blue curve) revealed that the intensities of Zn and O peaks increased. In contrast, the intensity of the C1 peak significantly decreased.

Enhancing Zn and O peak intensities is expected, as surface cleaning removes accidental contamination and exposes the “true” surface of the Zn-O film. Ultimately, the measured bulk composition comprised approximately 47% oxygen, 46% zinc, and 7% carbon. This indicates that carbon has a relatively low affinity for zinc, but some carbonaceous species from the working gas atmosphere remain trapped within the growing ZnO structure.

High-resolution Zn 2p3/2, O 1s, and C 1s core electron spectra from the non-sputtered Zn-O sample are presented in Figure 7. The Zn 2p3/2 peak was fitted by one component whose peak energy was located at 1021.1 eV. Later energy indicates that during deposition, nearly all zinc was fully oxidized to the Zn (II) state. This assumption is further supported by the main component of the O 1s peak whose binding energy is 529.8 eV as this value fits well with the reported binding energies for ZnO. All other solid peaks and their constituents (namely, the second O 1s component at approximately 531.5 eV and the C 1s components at approximately 284.8 eV, 286.0 eV, and 288.8 eV) can be associated with the adventitious contamination by hydrocarbons and adsorbed moisture. However, considering that “bulk” XPS results indicated carbon presence inside the film, it can be assumed that fractions of C-O (286.0 eV) and C=O (288.8 eV) bonds are not related to adventitious contamination but could be caused by the carbon located inside ZnO matrix.

### 3.3. Antimicrobial Activity of Textile with One-Sided and Two-Sided ZnO Nanoparticles against 16 Microorganisms

The antibacterial activity of textile samples with one-sided and two-sided-decorated ZnO 150 W/200 W was first checked against *Klebsiella pneumoniae* (*K. pneumoniae*) and *Staphylococcus aureus (S. aureus)*. The study results show that cotton fabric single-sided or two-sided-coated with ZnO 150 W/200 W nanoparticles had very similar antibacterial effects on *K. pneumoniae*; however, one-sided-coated with ZnO 200 W material inhibited better than two-sided-coated ZnO 150 W material (*p* = 0.1865). The opposite results were for *S. aureus*, as ZnO nanoparticles did not affect the growth of *S. aureus* when a cotton material was one-sided-coated with ZnO 200 W. Still, when the material was two-sided-coated, it inhibited bacteria growth by nearly 17% compared with the control material (*p* < 0.001) (Figure 8).

Next, the antimicrobial activity of a two-sided cotton material coated with ZnO 150 W was evaluated against 16 reference and clinical/zoonotic strains—bacteria (15 strains) and fungus (1 strain). ZnO 150 W is more effective against Gram-negative bacteria than Gram-positive (*p* < 0.001). Textiles treated with nanoparticles ZnO 150 W showed better antifungal activity against *C. albicans* than control fabric (*p* < 0.001). Thus, the results indicated that ZnO nanoparticle-coated fabric has antimicrobial efficiency (Figure 9).

SEM micrograph images of Gram-positive bacteria (*S. aureus*), Gram-negative bacteria (*K. pneumoniae*), and a fungus (*Candida tropicalis*) that were transferred onto glass without (control) and with ZnO nanoparticles 200 W are presented in Figure 10. ZnO nanoparticles caused cell damage—the *leakage* of the *cytoplasmic* content, the *cell membrane disruption, and deformation*. This can be attributed to the adsorption of nanoparticles, followed by penetration inside the cell and cell membrane damage [37].

Our previous work demonstrated that two-sided-coated cotton fabric with Fe_2_O_3_ nanoparticles has higher antimicrobial efficiency than one-sided-coated fabric [36]. Additionally, our research proved that cotton fabric treated on two sides with ZnO nanoparticles shows better antimicrobial properties than fabric coated on a single side with metal oxide nanoparticles. Better antibacterial action of cotton fabric two-sided-treated with ZnO nanoparticles was found against Gram-negative bacteria than Gram-positive, except *S. haemolyticus* and *S. epidermidis*. It does not confirm the newest results of the study performed by Riahi et al., 2023, as zinc oxide nanoparticles (ZnO NPs) synthesized by a sol–gel method were offered as a potential solution against various resistant Gram-negative, Gram-positive, or multidrug-resistant (MDR) bacteria [38].

It is worth noting that the nanoparticles’ impact on bacterial growth with different frequencies also varied depending on the nanoparticle’s size and concentration and the species of bacteria [39].

Additionally, the antimicrobial activity of ZnO might vary according to the different light conditions. Under dark conditions, ZnO action on microorganisms is due to mechanical contact with cell membranes as it was revealed that antimicrobial Zn^2+^ ions are released [40]. When light is present, ZnO nanoparticles can produce reactive oxygen species (ROS). Oxidative stress will be induced on the microorganisms, and the cell membrane experience damage [41], leading to the leakage of cell contents and cell death [42]. It was shown that biofilm formation by Gram-positive and Gram-negative can be interfered with by ZnO coatings [43]. Zinc oxide nanoparticles’ primary mechanism of action (MOA) has not been fully elucidated [44]. The antimycotic activity of ZnO is not that well investigated [45]. It is only known that ZnO nanoparticles can inhibit the growth of clinical isolates of *Microsporum canis* [46] and possess antifungal action against *C. albicans* in a dose-dependent manner [47]. Our study confirmed that *C. albicans* growth is strongly inhibited by a cotton material modified by ZnO nanoparticles, by nearly 60 percent. The antifungal mechanism of ZnO is described as an interaction of ZnO NPs with fungal cells, causing alterations in cell membrane integrity that might result in cell death [47].

## 4. Conclusions

Surface morphology, topography, and elemental and structural analyses confirmed the successful deposition of ZnO nanoparticles on textile surfaces. Green chemistry technology with low-temperature oxygen plasma was used to depose ZnO NPs. Cotton textile treated on two sides with ZnO nanoparticles demonstrated higher antimicrobial activity against Gram-negative bacteria and two types of Gram-positive bacteria, *S. haemolyticus* and *S. epidermidis*, compared to one-sided-coated cotton material. The obtained results indicate a high potential for producing antimicrobial nanotextiles. Future investigations might focus on research on ZnO photocatalytic action. The photocatalytic activity of ZnO-coated textiles absorbs a more significant fraction of the solar spectrum. Thus, these materials could reach higher antimicrobial efficiency against microorganisms.

## Figures and Tables

**Figure 1 nanomaterials-14-01264-f001:**
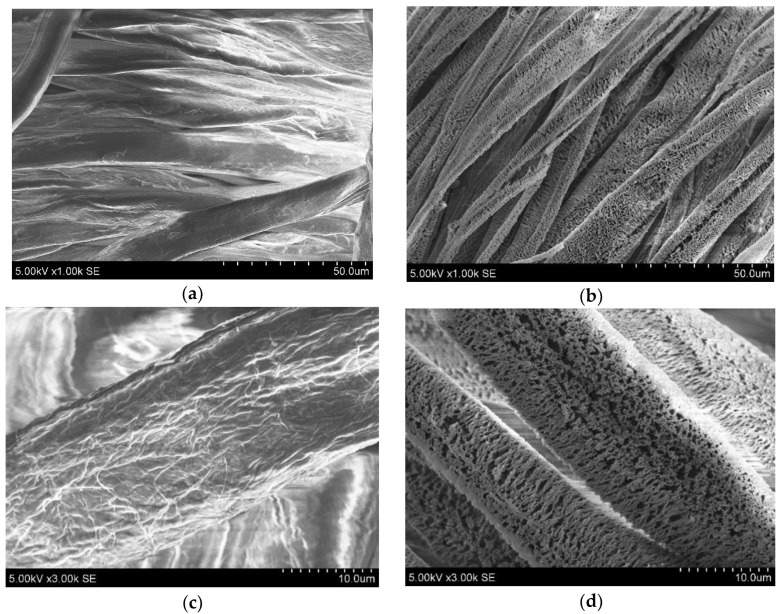
Textile surface morphology images of uncoated (**a**,**c**) and ZnO-coated surfaces (**b**,**d**) at different magnifications (1.00k and 3.00k, respectively).

**Figure 2 nanomaterials-14-01264-f002:**
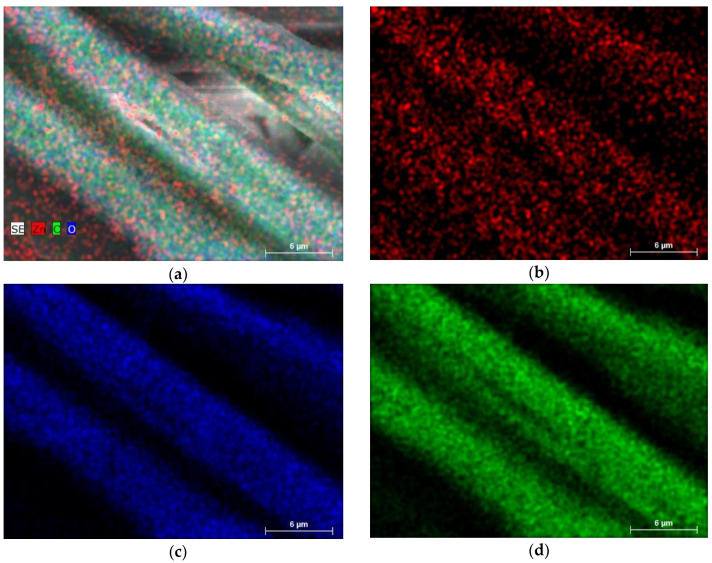
The elemental mapping images of all C, Zn, and O together (**a**) and individual elements Zn (**b**), element O (**c**), and element C (**d**), deposited on the textile surface.

**Figure 3 nanomaterials-14-01264-f003:**
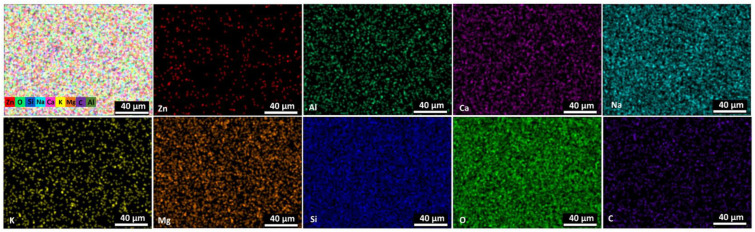
The elemental mapping images of Zn, O, Si, Na, Ca, K, Mg, C, Al as all elements and individual elements of Zn, Al, Ca, Na, K, Mg, Si, O, C on glass substrates.

**Figure 4 nanomaterials-14-01264-f004:**
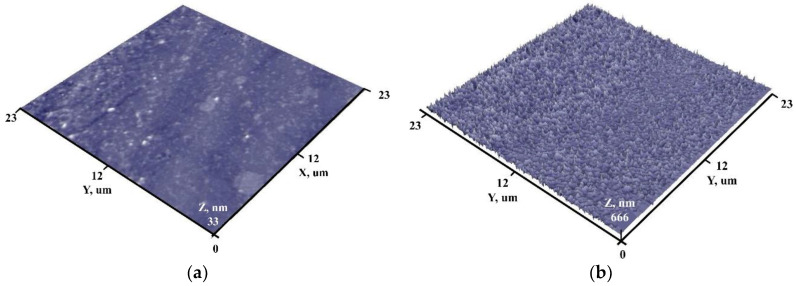
AFM surface views of initial surface (**a**) and with ZnO NP’s (**b**).

**Figure 5 nanomaterials-14-01264-f005:**
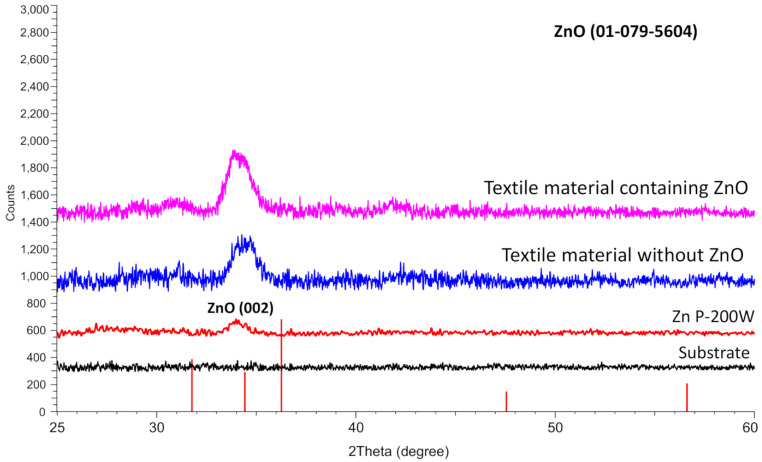
The XRD pattern of ZnO was deposited on glass slides (substrate without ZnO and glass substrate with ZnO deposited at 200 W plasma power) and textile material (textile material without ZnO and Textile material containing ZnO deposited at 200 W plasma power).

**Figure 6 nanomaterials-14-01264-f006:**
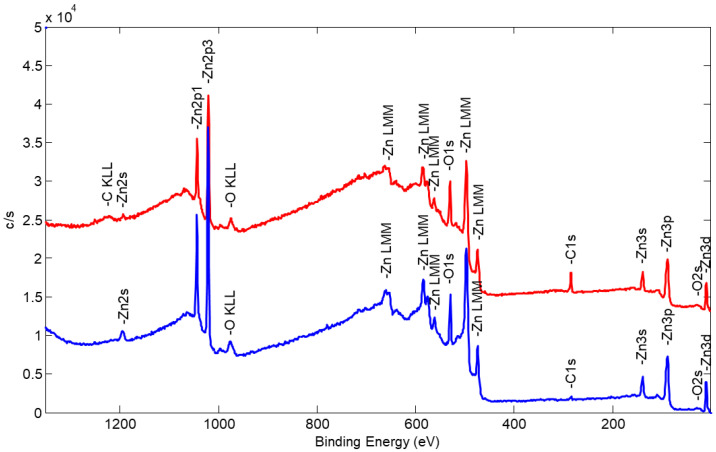
XPS survey spectra of Zn-O NPs deposited on a glass substrate: red—glass surface ZnO NPs as deposited, blue—after Ar^+^ ion gun sputtering.

**Figure 7 nanomaterials-14-01264-f007:**
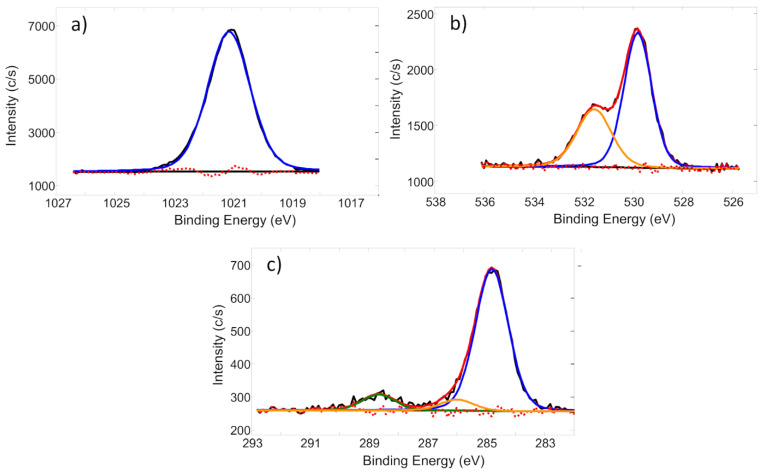
The high-resolution core-level photoelectron spectra: Zn 2p3/2 (**a**), O 1s (**b**), and C 1s (**c**) (The black curve represents the experimentally measured spectra, the red dashed line corresponds to the background signal, and the blue, orange, and green curves indicate the fitted spectra components).

**Figure 8 nanomaterials-14-01264-f008:**
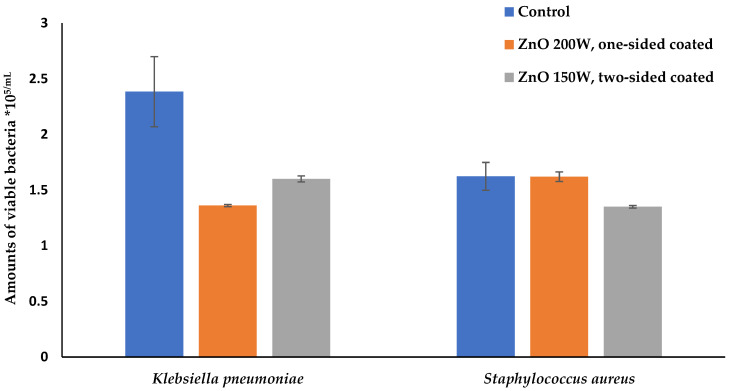
Amounts of viable bacteria *S. aureus* ATCC 2593 and *K. pneumoniae* ATCC 10031 cells (CFU × 10^5^/mL) after contact with uncoated (control), and one-sided and two-sided-coated with ZnO nanoparticle cotton material.

**Figure 9 nanomaterials-14-01264-f009:**
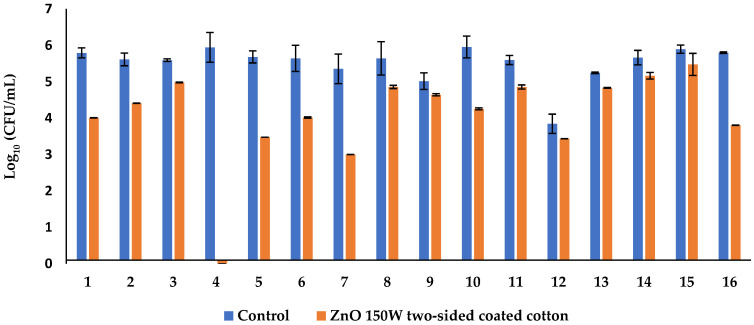
Bar graph displaying a logarithmic scale of bacterial growth of 16 strains before (control) and after contact with two-sided-decorated (ZnO NP’s) textile. (1) *Enterobacter cloacae*, (2) *Salmonella enterica*, (3) *Citrobacter freundii*, (4) *Aeromonas hydrophila* DSM 112649, (5) *Acinetobacter baumannii*, (6) *Pseudomonas aeruginosa* ATCC 27853, (7) *Staphylococcus haemolyticus*, (8) *Enterococcus faecalis* ATCC 29212, (9) *Enterococcus faecium*, (10) *Klebsiella pneumoniae* ATCC 10031, (11) *Staphylococcus aureus* ATCC 25923, (12) *Candida albicans* ATCC 10231, (13) *Corynebacterium* spp., (14) *Cutibacterium acnes*, (15) *Staphylococcus hominis*, and (16) *Staphylococcus epidermidis*.

**Figure 10 nanomaterials-14-01264-f010:**
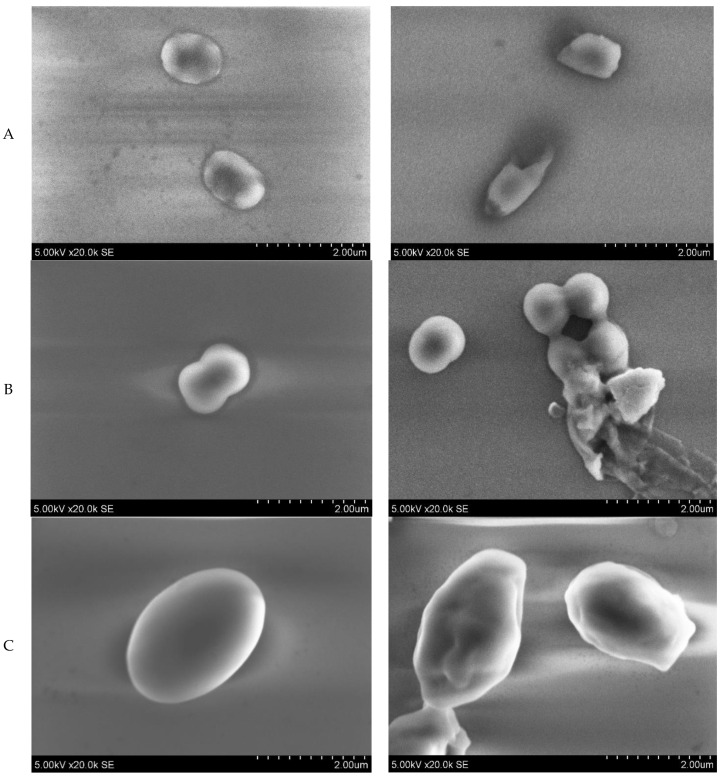
SEM micrograph images demonstrate non-treated (**left**) and treated with ZnO nanoparticles (**right**) bacteria *K. pneumoniae* ATCC 10031 (**A**), S. aureus ATCC 25923 (**B**), and a fungus *Candida albicans* (**C**).

**Table 1 nanomaterials-14-01264-t001:** Elemental composition analysis of textile surfaces using EDS.

Sample	Elemental Composition of the Textile Surface		
	C [at. %]	O [at. %]	Zn [at. %]
Before deposition	44.03	55.97	-
After ZnO NP deposition	47.25	51.16	1.59

**Table 2 nanomaterials-14-01264-t002:** Elemental composition analysis of glass surfaces using EDS.

Sample	Elemental Composition Analysis of Glass Surfaces, at. %								
	Zn	O	Si	Ca	K	Mg	Al	Na	C
Before deposition	0	60.28	23.90	2.52	0.41	2.27	0.48	9.97	0.18
After ZnO NP deposition	1.34	58.46	24.41	2.73	0.43	2.15	0.50	9.68	0.31

## Data Availability

The data presented in this study are available upon request from the corresponding author. The data are not publicly available, due to the next work.

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
