# Peer review of "ZnO Nanoparticles Enhance the Antimicrobial Properties of Two-Sided-Coated Cotton Textile"

_nanomaterials, 2024, doi:10.3390/nano14151264_

Round 1
Reviewer 1 Report (Previous Reviewer 2)
Comments and Suggestions for Authors
I am pleased with the authors' responses to my previous comments. Their explanations and revisions have been satisfactory.
Author Response
Dear Reviewer,
Thank you very much for an evaluation and review of the manuscript „ZnO Nanoparticles Enhance Antimicrobial Properties of Two-sided Coated Cotton Textile“. Your valuable comments on the article helped us to improve it and interpret the results/findings of the study more accurately and concisely.
Sincerely,
The Authors
Reviewer 2 Report (Previous Reviewer 3)
Comments and Suggestions for Authors
1. The vertical axis of Fig.9 is not clear,
2. From Fig.8 and Fig.9, it can be seen that the antibacterial effect is not very significant compared to the controls. Why is the effect of the one-sided better than the two-sided, and it is difficult to explain convincingly?
3. Why choose glass substrate as a control and whether to remove related content?
Author Response
Dear Reviewer,
Please find the response to the comments.
Comments 1: The vertical axis of Fig.9 is not clear,
Response 1: We agree with the comment. The name of the vertical axis “Log10 (CFU/mL) “ (Figure 9) is overlapping due to technical issues with the main picture. Figure 9 was updated to a new version and placed in the manuscript.
Comments 2: From Fig.8 and Fig.9, it can be seen that the antibacterial effect is not very significant compared to the controls. Why is the effect of the one-sided better than the two-sided, and it is difficult to explain convincingly?
Response 2: Our research showed that cotton fabric treated on two sides with ZnO nanoparticles had better antimicrobial properties than fabric coated on a single side with metal oxide nanoparticles. However, more detailed studies are needed to show the significant differences in the effects of one-side and two-side coated cotton material as an antimicrobial action of metal nanoparticle-coated fabrics depends on various factors such as the type of metal, metal nanoparticle's size, concentration/dose on the material, deposition time, and the species of bacteria.
Comments 3: Why choose glass substrate as a control and whether to remove related content?
Response 3: The damage to the microorganism (a leakage of cytoplasmic content, cell membrane disruption, and deformation) after contact with ZnO particles was evaluated using scanning electron microscopy. Better photo quality is obtained if ZnO nanoparticles are deposited onto the surface of microscope glass slides than on the cotton material samples.
Sincerely,
The Authors
This manuscript is a resubmission of an earlier submission. The following is a list of the peer review reports and author responses from that submission.
Round 1
Reviewer 1 Report
Comments and Suggestions for Authors
I don't have any major comments about the work. However, some improvements are needed for the drawings and captions:
- Figure 2: It would be beneficial to clarify the meaning of the colors in the pictures. Consider adding this information to the description of the drawing.
- Figure 4: The scale on the axes is currently not visible. It needs to be made more prominent for better readability.
- Figure 5: Consider adjusting the range on the y-axis to a smaller value. This adjustment will enhance the visibility of the ZnO peak.
- Figure 7: The scales on both the x and y axes are not clearly visible. Ensure they are clearly marked for better interpretation.
After making these corrections to the figures, the article will be ready for publication.
Reviewer 2 Report
Comments and Suggestions for Authors
The article "ZnO nanoparticles enhance antimicrobial properties of two-sided coated cotton textile" describes the obtaining of cotton fabric with deposited ZnO nanoparticles, characterization of ZnO thin film on a glass substrate and antimicrobial activity of the cotton coated with ZnO nanoparticles. It is a valuable study that can be published after authors address the following problems:
Abstract should be checked and revised carefully by briefly introducing the work plan and key findings. Abstracts should highlight the innovation of the article, as often abstract section is presented separately in search engines, it must be able to stand alone as an informative piece. In the abstract, need to focus more on the quantitative information, not qualitative one. The key quantitative data showing the antimicrobial efficiency should be included in the abstract (after authors clarify what they measured exactly).
The English language needs some polishing for style and typos (e.g. Gram-positive/negative should be with capital G as it comes from a person name- rows 43-44 and elsewhere; rows 163-164 are about uniformity or non-uniformity for Zn in EDS?; rows 195-196 must be rephrased; all Latin names like K. pneumoniae should be italicized – row 234 and further; legend of figure 9 has different font size; authors should evaluate when to use antimicrobial and when antibacterial in the text – tests made only on bacteria are better to be referred as antibacterial – figure 8, while test made on bacteria and fungi can be referred as antimicrobial – figure 9; explain the term positive nanoparticles from row 286; use uniform style in reference section – some names are randomly underlined.
Authors made a starting assumption that the ZnO will deposit in the same fashion on cellulose fibres and on glass substrate, and this must be supported by other literature reports.
The peak corresponding to the plan with Miller indices (002) is not the most intense in the XRD of ordinary ZnO nanopowder. The fact that is the only peak observable indicate a preferential growth of ZnO crystals after this direction, which must be correlated by authors with the glass structure (SiO2). This is an important find that also must be correlated with the previous observation that ZnO might not grow in the same fashion on cellulose and glass. On glass surface the ZnO preferentially growth on (002) please see doi: 10.1016/j.heliyon.2017.e00285 or doi: 10.3390/app9214509, and is up for authors to explain the mechanisms. At the same time the ZnO deposited on cellulose has the usual XRD pattern with all the peaks, see doi: 10.1007/s10570-011-9523-1; doi: 10.1007/s10570-018-1996-8; doi: 10.1016/j.carbpol.2017.02.020. Again is up to authors to convince us that ZnO would crystalize in the same fashion on this two different substrates, otherwise they cannot claim that by characterizing ZnO deposited on glass, the results are good also for ZnO deposited on cellulose.
One fundamental issue that authors should address is the strong photocatalytic activity of ZnO. As cotton fibres are usually dyed and cloths are wear outside, the presence of ZnO NPs can degrade the organic dyes very quickly (see doi: 10.1016/j.ceramint.2022.11.178; doi: 10.3390/ijms24065677). As such the authors should propose a way to decrease photocatalytic activity (decrease ROS production) of ZnO NPs, but this will impair on their antimicrobial activity, as ROS production is one of the several mechanisms involved. A second pathway is that such treated textile to be used only in white fabrics, underwear, wound dressing etc.
At row 187 authors are talking about decreasing of surface roughness after ZnO deposition? The figure 4b and the Ra value indicate the opposite.
This work is interesting and can be boosted further. Hence the following literature could prove this manuscript doi: 10.1016/j.arabjc.2018.12.003; doi: 10.3390/pharmaceutics14122842 and help depict the possible killing mechanisms.
Figure 8 – the results of antibacterial activity – is an uncommon fashion to present the potency of any antimicrobial agent. The usual way, which also helps comparison with other literature results is as chart-bars or table with numerical values. Like in any biological test, error bars, SD and significant differences should be presented. The results without statistical interpretation are meaningless for biological systems. Additionally, it is not clear what did authors measured: OD, CFU, MIC etc? Please see doi: 10.1039/C3RA41231H for clarity.
P values are presented differently at rows 255 and 256 (base 10 or e). Again figure 9 is hard to evaluate, like which is the difference among 13,14 and 17? That is why logarithmically scales are usually presented in such graphs.
The conclusion should reflect the heuristic of the study. How is this system a better one vs other antimicrobial textile, treated with metallic or oxide nanoparticles? Conclusion section must be reworked to underline the novelty and advantages of this research, with actual numbers. The conclusion part does not highlight the salient findings and future perspective (please see the observation on the photocatalytic activity of ZnO).
Reviewer 3 Report
Comments and Suggestions for Authors
1. The antibacterial activity of heavy metals and their oxides is widely known, and even the antibacterial mechanism has been extensively studied. The author's description in the title, abstract, and introduction does not fully demonstrate the importance and innovation of this study.
2. The term "nanomaterials derived from cotton" in the abstract is ambiguous and may be mistaken for nanomaterials derived from cotton.
3. What is the significance of using glass slide for another coating? Why not conduct surface testing and analysis directly on the fabric (fiber)? Will the significant differences in morphology, structure, and chemical composition between the Glass surface and the Cotton surface result in different deposition effects?
4. Figure 5 should be reprocessed or resampled for testing.
5. Can the coating be said to be a nanoparticle or a metal (oxide) film or membrane)?
6. The fastness of ZnO coating should be given, such as water washing resistance, friction resistance, etc.
7. How to coordinate the sizing and dyeing process of Cotton fabric (clothing) with the coating of this research?
Comments on the Quality of English Language
No.